# Exploring the Inclusion of Sustainability into Strategy and Management Control Systems in Peruvian Manufacturing Enterprises

**Luis Jesús Córdova-Aguirre** [1] and **Juan Manuel Ramón-Jerónimo** [2,*]

1   Faculty of Engineering and Architecture, University of Lima, Av. Javier Prado Este 4600, Santiago de Surco, Lima 15023, Peru; lcordova@ulima.edu.pe
2   Department of Financial Economics and Accounting, University Pablo of Olavide of Seville, Ctra. Utrera, Km. 1, 41013 Sevilla, Spain
*   Correspondence: jmramjer@upo.es; Tel.: +34-954-349-177

**Abstract:** The aim of this work is to explore the incorporation of sustainability into strategy and management control systems (MCSs) in Peruvian manufacturing enterprises in the plastics sector. The study focuses on identifying and analyzing the current way they incorporate and manage sustainability to determine the shortcomings that must be corrected in the future to design an effective performance management system (PMS) that includes sustainability to help companies achieve sustainable growth. The method of multiple case study analysis was used. Data was obtained from four Peruvian manufacturing firms in the plastics sector through seven semi-structured interviews. The findings suggest that sustainability is partially incorporated into the company's strategy, and that especially in medium-sized enterprises, managers do not know how to implement sustainable management accurately. Concerning MCSs used, in medium-sized companies, short-term planning is carried out and they are limited to the control of the economic operational perspective and lack concrete measures regarding social and environmental aspects. Finally, we conclude that this study allowed us to know how sustainability is really managed in Peruvian manufacturing enterprises in the plastics industry and that it is necessary for these companies not only to incorporate sustainability into their strategy but also to implement a holistic PMS to be used as a broad-scope MCS to achieve sustainable growth.

**Keywords:** sustainability; triple bottom line; strategy; management control system; performance management system; balanced scorecard

## 1. Introduction

According to the Human Development Report 2020 [1], the Earth has entered an entirely new geological epoch: the Anthropocene, or the age of humans. It means that we are living in an age defined by human choice, in which the dominant risk to our survival is ourselves. In that sense, "we need a just transformation in the way we live, work, and cooperate" [1] (p. 2). Working with—not against—nature and society is needed. It is time to make a change. "Our future is not a question of choosing between people or trees; it is neither or both" [1] (p. 3).

To achieve the above mentioned, the future of the Earth depends entirely on the actions of man, but those actions will depend on the choices he makes in every area or organization. One of the main entities whose actions affect the climate and society are manufacturing companies. However, the decisions made in these companies will depend on the knowledge and conviction the decision-makers have about certain aspects on which they will base them. Therefore, it is highly necessary to know what businessmen understand about sustainability and whether this concept is included in the strategies and management control systems of the companies they manage.

In addition, it should be mentioned that it is crucial for organizations to incorporate sustainability into their strategy for generating a competitive advantage [2–4]. However, the integration of sustainability into corporate strategies has not yet been fully resolved and the literature indicates that more research is needed in this area, especially in emerging economies in general [5], as is the case of Peru.

To integrate sustainability into the strategy, organizations in developed countries are using business models that incorporate the triple bottom line (TBL) approach [6]. However, although many organizations work with the three dimensions of TBL, they do not explore their interconnections [7,8]. In this sense, some authors [9] proposed using the balanced scorecard (BSC) to do so.

In this research, we explore and analyze how sustainability is currently incorporated into the strategy and how it is managed and controlled in Peruvian manufacturing companies in the plastics industry, so that the findings can serve as a basis for the subsequent design and development of a holistic PMS integrating the aspects of TBL into the strategy to be managed through BSC perspectives.

The study was carried out in manufacturing companies due to the important role they play in the economy and the progress of countries through the creation of jobs [10]. Furthermore, companies in this sector use a great number of resources in their production processes, so their impact on the three dimensions of the triple bottom line approach is significant.

Within the manufacturing sector, it was decided to study the plastics industry because it is an important industry whose products are used by all industries and whose demand has only grown since its discovery. In Peru, the plastics manufacturing industry contributes to 4% of industrial GDP, and generates 52,000 direct jobs and 13% of the industry's internal taxes [11].

Likewise, it was decided to carry out this research in Peru because in this country there is little awareness of the need to adopt sustainability in companies. Only 5% of Peruvian companies apply social responsibility management [12], and although some companies use social and environmental indicators, they are used in isolation and only for voluntary informative purposes to report to governmental entities and stakeholders.

Lastly, it should be noted that the work's value lies in the fact that the study was carried out in a country with an emerging economy, where only a minority of companies apply corporate social responsibility management, and in an industry that has scarcely been addressed in previous studies.

The rest of the paper is organized as follows: In Section 2, a literature review and a presentation of some related theoretical concepts are included to support the proposed conceptual framework. In Section 3, the research methodology is described, together with data collection strategies. Empirical findings are presented in Section 4. In Section 5, a discussion and the conclusions of the study are provided, including research limitations and further research lines.

## 2. Theoretical Development

### 2.1. Brief Industry Description

The Peruvian manufacturing sector is a driver for growth, development, and employment. According to the Institute of Economic and Social Studies (IEES) of the National Society of Industries (SNI) of Peru, this sector contributed USD 4.138 billion to the national treasury in 2018, a figure that represents 15.9% of domestic revenue. With this, it was consolidated for the sixth consecutive year as the first economic activity in the country with the highest contribution in taxes. The industrial branches with the highest contribution were food and beverages, chemical products, rubber, and plastics, and metal products, machinery and equipment [13]. Besides, in 2019, manufacturing activity employed 1,519,170 people, representing 8.9% of the Peruvian employed EAP [14].

On the other hand, the plastics industry contributes to 4% of the industrial GDP; generates a total of 200,000 jobs [15], of which 52,000 are direct jobs; generates 13% of the

industry's internal taxes; represents 4% of non-traditional exports; and is the destination of 7% of the credit directed to manufacturing [12].

According to [16], over the past six years, the Peruvian manufacturing of plastic products (ISIC 2220—Revision 4) has sustained periods of growth and contraction. In 2013 and 2018, the production of plastic products expanded 11.2% and grew at an average annual rate of 2.2%. In 2018, the industrial production of plastic products expanded by 4.5%, the highest growth rate since 2014. This good performance was due to the increased demand for plastic products for construction sites, transportation items, packaging, and containers to meet domestic demand. Likewise, external demand also drove the higher production of plastics by growing 16.0% compared to what was recorded in 2017. In the first four months of 2019, the industrial production of plastic products grew 4.2% compared to the same period in the previous year. Between January and May 2019, Peruvian exports of manufactured and semi-manufactured plastic products grew 3.6% compared to the same period in 2018, reaching USD 189.1 million. Likewise, in terms of volume, foreign sales totaled 68,749 tons, a volume that meant a year-on-year growth of 5.6%. In relation to the main trading partners, it is worth highlighting sales to the United States, a country that represents 16.0% of the total value exported.

As we know, plastic is a highly versatile product due to the multiple shapes and textures it can take. This characteristic has made it one of the most widely used raw materials in different applications worldwide: the automotive industry, construction and building, medicine, electrical and electronics, food, and agriculture, among other sectors. Despite these benefits, in recent years the plastics industry around the world has faced pressure against its productive activity, mainly against the manufacture of single-use plastics, since many of them are thrown into rivers and seas by the population, negatively impacting the environment. This fact has led governments in different parts of the world to implement measures to limit the use of plastic, ranging from the establishment of taxes on consumption and/or commercialization, to the partial or total prohibition of the use of plastic [16]. In that sense, during the II International Congress of the Plastics Industry held at the headquarters of the National Society of Industries of Peru, the president of the guild mentioned that the need to achieve a sustainable plastics industry must be promoted: "We must generate a circular economy, rethink what we are doing, redesign our products, reduce plastic consumption as consumers, and convert toxic products back into raw materials. We must work on this in schools and universities. More recycling, more life, is the motto of the SNI" [15]. However, it should be mentioned that only 5% of Peruvian companies apply social responsibility management [12], and although there is an environmental control legislation for certain products in the plastics manufacturing sector (prohibition of the use of single-use plastics, the use of non-biodegradable materials for bags, and the use of expanded polystyrene for food and beverage containers for human consumption), the law 30,884 came into force in December 2018, but companies have until December 2021 to comply.

*2.2. Literature Review*

It is crucial for organizations to incorporate sustainability into their strategy for generating a competitive advantage [2–4]. However, the integration of sustainability into corporate strategies has not yet been fully resolved. Moreover, the literature indicates that "more research is needed on dynamic capabilities for the achievement of sustainability, especially in emerging economies in general" [5].

To integrate sustainability into the strategy, organizations in developed countries are using business models that incorporate the triple bottom line (TBL) approach [6] that was created by John Elkington in 1997, in his book, "Cannibals with forks: The Triple Bottom Line of 21st Century Business" [17,18], taking the Bruntland Commission's [19] approach to sustainable development and adapting it to the business environment. In it, he takes up the idea of a three-dimensional vision of development and coins the term "triple bottom line" to express the idea that it is necessary to provide a framework for measuring

business performance and the success of an organization based on the economic, social, and environmental dimensions. However, although many organizations work with the three dimensions of the TBL, they do not explore their interconnections [7,8]. In this sense, Kalender and Vayvay [9] proposed using the balanced scorecard (BSC) "as a bridge to implement sustainable strategy and link corporate sustainable objectives with actions and performance results" (p. 76).

Since its creation, the BSC has been evolving and even the management control literature has already analyzed models that include the social and environmental perspective to the traditional BSC and indicators have been proposed [3,4,9,20–29]. In this sense, the BSC has a high potential to integrate social and environmental aspects into the overall management system. Besides, using the BSC approach based on sustainable development parameters is a powerful and useful methodology to evaluate the sustainable performance of the company or organization [9].

Complementarily, it is stated that more extensive research and collaboration is needed to improve understanding of sustainability in manufacturing [30]. The manufacturing sector is an important actor in achieving sustainable development. Its role in creating jobs, improving social welfare, and reducing environmental impact has led to increased attention for research in the field of sustainable manufacturing [10]. Manufacturing activities consume a large amount of energy and natural resources [31,32] and produce more emissions to air and land, while having significant implications for the society and economy. In that sense, the control of and reduction in environmental and social impacts have become an additional objective in manufacturing industries [31,33,34]. However, even though sustainability assessment is a crucial means to promote sustainable development, relatively few methodologies and tools are applied in the manufacturing environment and they lack a holistic approach to sustainability [35]. There is a considerable lack of concrete measurements, efficiency analysis, and comprehensive controlling tools that encompass economic, social, and ecological issues [36].

Challenges for contemporary controlling practices are to link the larger variety of sustainability-oriented strategic goals to operations, to measure sustainability, and to evaluate the achievement of corporate goals [37–42]. Studies reveal that both controlling practices and sustainability are not yet routine and commonplace but differ significantly at the company and the country level [42,43]. Controllers are more involved in short-term operational tasks and functions than in long-term strategic issues [44].

In order to provide a perspective more focused on the operation of overall control systems looking beyond the measurement of performance to the management of performance, Ferreira and Otley [45] proposed an extended PMS framework that considers the following items: vision and mission, key success factors, organization structure, strategies and plans, key performance measures, target setting, performance evaluation, reward systems, information flows, and system and networks, all of them associated with the contextual factors and the organizational culture.

### 2.3. Proposed Conceptual Framework

The research question posed in this work is:

How could sustainability be effectively incorporated into the strategy and the MCSs of Peruvian manufacturing enterprises?

Hence, to answer that research question, and according to what is cited in the literature review and in the sense that adopting a broad-scope design of MCS is necessary to improve long-term performance [46–48], we propose a holistic performance management system (PMS) (Figure 1) to assist Peruvian industrial managers to effectively incorporate sustainability in the strategy and the management control systems of the companies they run. The proposed system embeds the three aspects of the TBL in the vision, mission, values, and strategic objectives of the company, and leverages the potential of the BSC to manage and control the accomplishment of the strategic objectives [49] to achieve economic benefits in a sustainable manner. This proposed conceptual framework also complies with

the theory of the four control levers proposed by Simons [50]: belief systems, boundary systems, diagnostic control systems, and interactive control systems. The first two levers concern the strategic formulation (vision, mission, values, and strategic objectives) and the other two levers concern the management and control of the strategy through the balanced scorecard. It should be noted that the concept of sustainability must be imbued in these four levers.

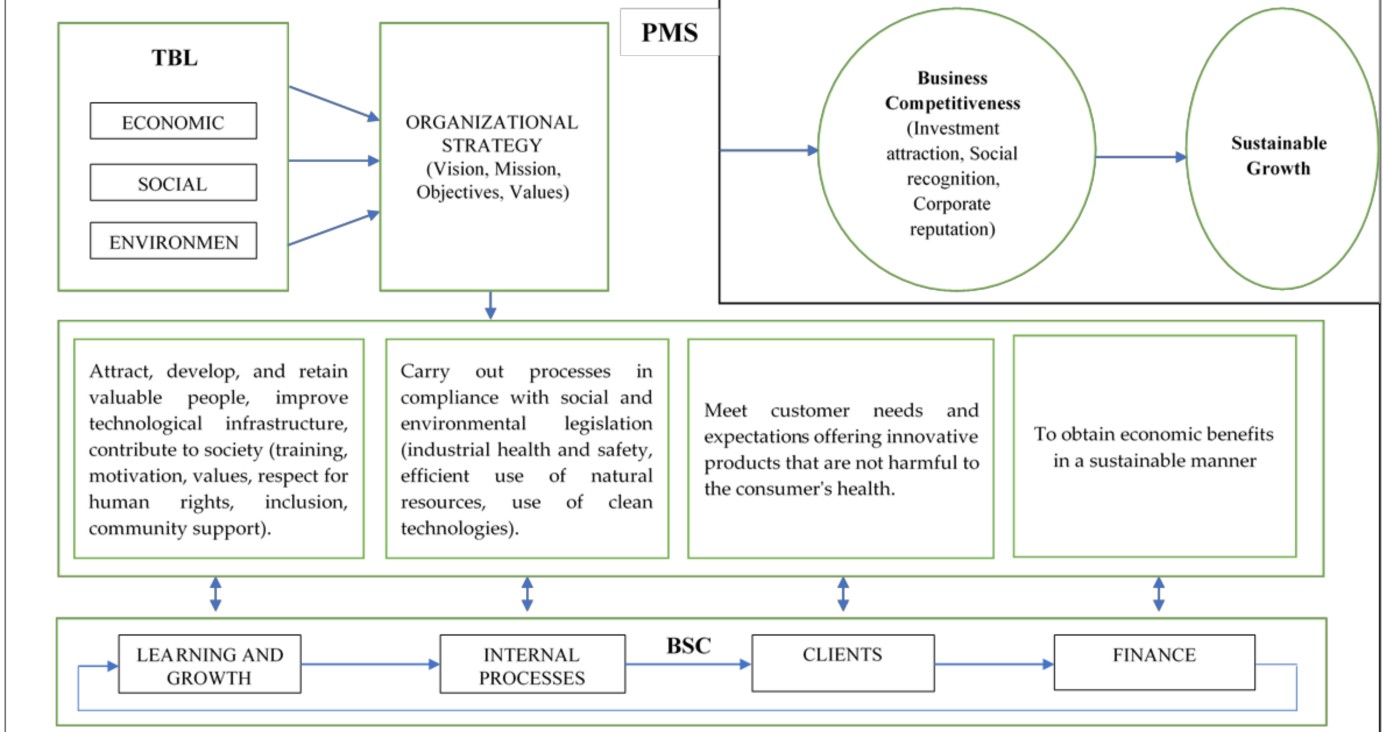

**Figure 1.** Proposed conceptual framework.

Finally, it should be mentioned that the proposed system requires that the strategic objectives seek to attract, develop, and retain valuable people; improve technological infrastructure; contribute to society (training, motivation, values, respect for human rights, inclusion, community support); carry out processes in compliance with social and environmental legislation (industrial health and safety, efficient use of natural resources, use of clean technologies); and meet customer needs and expectations, offering innovative products that are not harmful to the consumer's health. Complementarily, appropriate management indicators must be developed to monitor the achievement of the strategic objectives in the short term.

## 3. Methodology

The purpose of the present study is to explore the incorporation of sustainability into strategy and management control systems in Peruvian manufacturing enterprises in the plastics sector. Specifically, the research seeks to establish whether Peruvian entrepreneurs understand the concept of sustainability and whether they are aware of its importance, and whether they are incorporating it in the strategy and in the management control system and the performance measurement system of their enterprises. Our research was classified as qualitative since we aimed to study the why and the how of the variables instead of the mathematical significance of the variables' relationships. Because of the explicative nature of this study, a multiple case study methodology was selected, which allows the researcher an in-deep analysis of the most relevant characteristics of a contemporary phenomenon within its real context [51].

### 3.1. Construction and Design of the Interview Questionnaire

Each semi-structured interview contained 24 open questions (see Appendix A). The interviews were structured as follows: personal data, company data, definitions and meaning of sustainability, strategic aspects, management control, and performance measurement. Each interview was video recorded, transcribed, and sent to the interviewee for validation. As a previous stage, a literature review was performed to define the interview script to be used in each interview and the questions were based on the five theoretical concepts mentioned in Section 2, especially on Ferreira and Otley's [45] extended performance measurement framework.

### 3.2. Data Collection

Data were obtained from four Peruvian manufacturing firms in the plastics sector (one large-sized enterprise and three medium-sized enterprises) through seven semi-structured interviews (Table 1), following Eisenhardt's recommendation to carry out between four and 10 cases [52].

**Table 1.** Interviews conducted.

| Enterprise | Interview | Period of Time | Interviewee Charge | Interviewee's Guild Position | Enterprise Size | Main Product Line | Main Production Proccess |
|---|---|---|---|---|---|---|---|
| A | 1 | 57′28″ | General Manager | Vice President of the Plastics Committee of the National Society of Industries of Peru | Medium | Thermoformed pacakges | Thermoforming molding |
| | 2 | 57′10″ | Chief of Production | | | | |
| | 3 | 41′18″ | Head of Administration and Finance | | | | |
| B | 4 | 45′49″ | General Manager | President of the Plastics Committee of the National Society of Industries of Peru | Large | Pipes and water tubes for water and sewage installation | Extrusion molding |
| C | 5 | 56′23″ | General Manager | | Medium | Bags and films | Extrusion and blow molding |
| D | 6 | 1 h 01′39″ | General Manager | | Medium | Articles for cleaning | Injection molding |
| | 7 | 1 h 03′02″ | Chief of Production | | | | |

It is important to mention that two of the seven interviews were conducted with the president and vice president of the Plastics Committee of the National Society of Industries of Peru, which is the highest governing body of the plastics industrialists' guild of Peru.

### 3.3. Data Analysis

To analyze the case data, the "explanation-building" interpretative technique was used [51]. Once each interview was transcribed, "fact sheets" in Microsoft Word were created for each of the studied variables. Each factsheet included two parts: The first one was called "key points" and contained the different characteristics observed in the specific respondent company in relation to each variable, and the second one was called "main trends," where the key points were further developed, including all the observed trends and singularities. Finally, every point included in the fact sheets was supported by several quotes from different interviews.

## 4. Findings

Following the previous process, the findings are presented according to the main variables identified in the study: sustainability and triple bottom line, strategy, and management control systems.

### 4.1. Sustainability and Triple Bottom Line

All the interviewees were familiar with the term sustainability and understood its importance. They were aware of the meaning of the term sustainability and stated that it was considered in the company's activities. "Of course, sustainability is the ability to meet our current needs without affecting those of a future generation. Sustainable development seeks a balance between economic growth, care for the environment and social welfare", one of the interviewees stated. Likewise, almost all interviewees understood the meaning of TBL and considered the practice of sustainability a priority factor for the growth of the plastics manufacturing sector, the company, society, and the country. "We are totally convinced that the industrial manufacturing activity and especially the plastics sector will only be sustainable if we work on the three dimensions of the triple bottom line", another interviewee claimed.

As for the medium-sized companies, they applied actions such as the use of recyclable materials for the environmental aspect and in the social aspect they focused on compliance with current labor legislation, but they do not currently use any system to measure these aspects. One medium-sized businessman stated that it is difficult to carry out these types of actions maybe because they do not know how to start.

"As a medium-sized company, it is very difficult to take this type of actions. In general, in Peru there is a lack of sincere commitment to ecology; we are not yet imbued with the meaning of TBL. It may also be that we do not know how to start. It is an issue that needs a lot of guidance from the government."

As for the large company, in addition to knowing the meaning of TBL, it applied concrete actions such as clean production voluntary agreements with the Ministry of the Environment and the Ministry of Production to improve internal processes focused on sustainable management and participation in contests for the creation of new products from recycled plastic. The company was also ISO9001, ISO14001, and ISO18001 certified.

Regarding the priority in the order of importance of the three aspects of the TBL, all the interviewees mentioned that the priority factor was the economic factor, followed by the social aspect and then the environmental aspect (Figure 2), mentioning that if there were no economic results, the rest could not be fulfilled. In that sense, one of the interviewees stated,

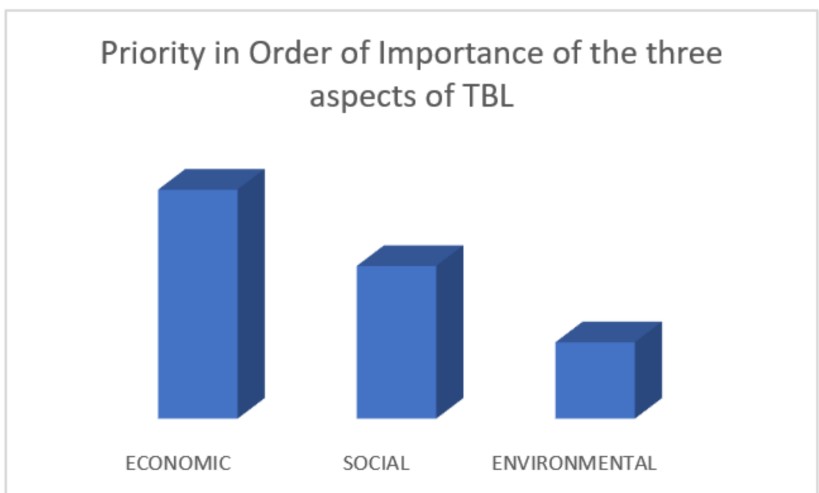

**Figure 2.** Priority in order of importance of the three aspects of the TBL.

> "The economic aspect is the most important, to the extent that if we are successful in the economic results, we will have resources to carry out social and environmental actions. To be transparent, sincere and realistic in order of priority, the importance would be: first the economic factor, then the social factor and then the environmental factor."

Social actions were only limited to the internal sphere of the company, specifically, to compliance with labor legislation, but there were no social actions towards the external sphere of the company, except in the large company. They believed that the accomplishment of labor legislations on social benefits and health and occupational safety was sufficient.

Most of the interviewees mentioned that they took social and environmental actions due to labor regulatory or economic reasons. No company reported its social or environmental actions on a mandatory basis. In this regard, one of the interviewees stated, "No, there is no agency that requires reporting of environmental and social actions on a mandatory basis, except for reports of compliance with occupational safety and health regulations, which are always reported to the Ministry of Labor, like any other company". As mentioned above, only social information in respect to compliance with labor legislation was reported, like any other company, regardless of whether it was in the plastics industry or not. In terms of environmental issues, there is no regulatory agency that requires any reporting. The large company stated that it voluntarily reports environmental improvements to the Ministry of the Environment to comply with the voluntary clean production agreement signed by the company. There is environmental control legislation for certain products in the plastics manufacturing sector (prohibition of the use of single-use plastics, the use of non-biodegradable materials for bags, and the use of expanded polystyrene for food and beverage containers for human consumption). Law 30,884 came into force in December 2018, but companies have until December 2021 to comply.

Regarding the environmental aspect, as mentioned above, all the interviewees placed it last in priority. It can be deduced that there are two main reasons why Peruvian businessmen put the environmental aspect as the last priority in importance. The first is that although there is a regulatory law on the use of certain plastic materials, it will be mandatory to fully comply with it only at the end of this year. The second reason, and in our opinion the most important, as one of the interviewees mentioned, is that most of the market is governed by the price of the products. In that sense, most of the clients of the manufacturing companies did not demand compliance with environmental and social protocols but rather the lowest price.

> "It depends on the type of client. There are clients who are interested in complying with these aspects, but there are clients who are not. Complying with all aspects is aimed at formal companies of a certain size. To comply with all aspects is costly, it costs money. The vast majority of Peruvian market is guided by price and is not interested in whether or not all aspects are complied with."

Consequently, the Peruvian businessmen were not fully concerned with complying with this aspect.

International and domestic large-sized clients require compliance with certain social responsibility protocols, such as the use of non-toxic materials, compliance with labor legislation, and compliance with technical standards in production processes; small and medium-sized clients, who represent most of the customer portfolio of these companies, do not.

### 4.2. Strategy

In general, with respect to strategic management, almost all medium-sized company interviewees mentioned that long-term strategic planning was not carried out in their companies. As for the large company, the interviewee mentioned that a long-term strategic plan was in place (Figure 3).

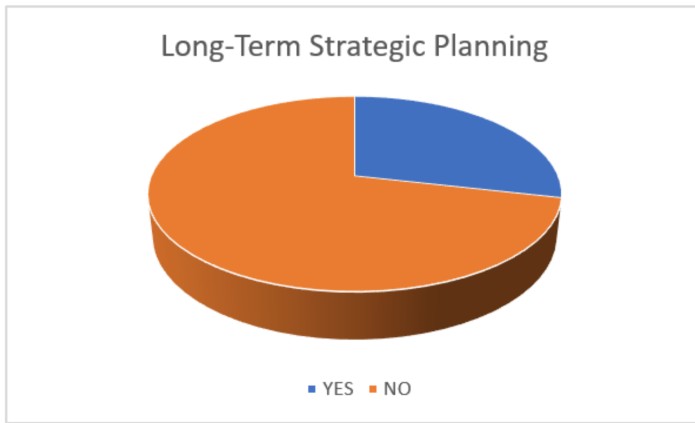

**Figure 3.** Use of long-term strategic planning.

All interviewees responded that the sustainability aspect must be considered in the strategy of a manufacturing company (Figure 4). The respondent who mentioned that they had a strategic plan stated that the sustainability aspect was incorporated into the strategic plan (vision, mission, values, strategic objectives) of their company. Those who did not have a strategic plan mentioned that, in general, the sustainability aspect was incorporated into their functional plans for the medium and short term.

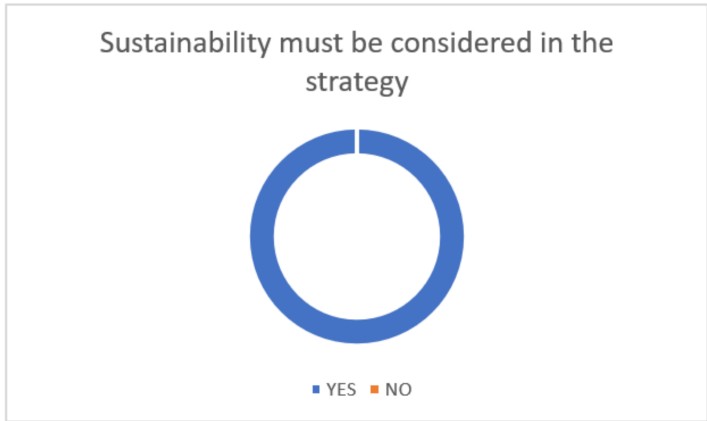

**Figure 4.** Consideration of sustainability in the strategy.

Only three of the interviewees, including the representative of the large-sized company, mentioned social and environmental aspects as one of the main key factors for the future success of their organization. The others did not mention it, mentioning only economic and commercial aspects.

Most of the interviewees believed that the management of social and environmental aspects positively influences the economic performance of a manufacturing company, and that it is important to consider them to achieve a competitive advantage. Only one interviewee had doubts about this, as was previously mentioned, stating that it depends on the market the company is targeting, because adopting social and environmental actions generates costs for the company and most of the Peruvian market is guided by price and is not interested in whether the company complies with all the aspects.

With respect to the stages of company greening [53], it could be said that the medium enterprises were in the stage of technology improvement and the large enterprise was just starting to embrace the green strategy.

*4.3. Management Control Systems*

As for MCSs, in medium-sized companies, these were limited to the control of the short-term operational perspective. Although they use tools to measure the company's per-

formance, they are often precarious and tailor-made and lack concrete measures regarding social and environmental aspects.

All the interviewees were familiar with the balanced scorecard, but only the interviewee from the large company used it to control and measure compliance with the strategic plan (Figure 5). Medium-sized companies used indicators by areas with customized software to control management and measure the performance of the areas.

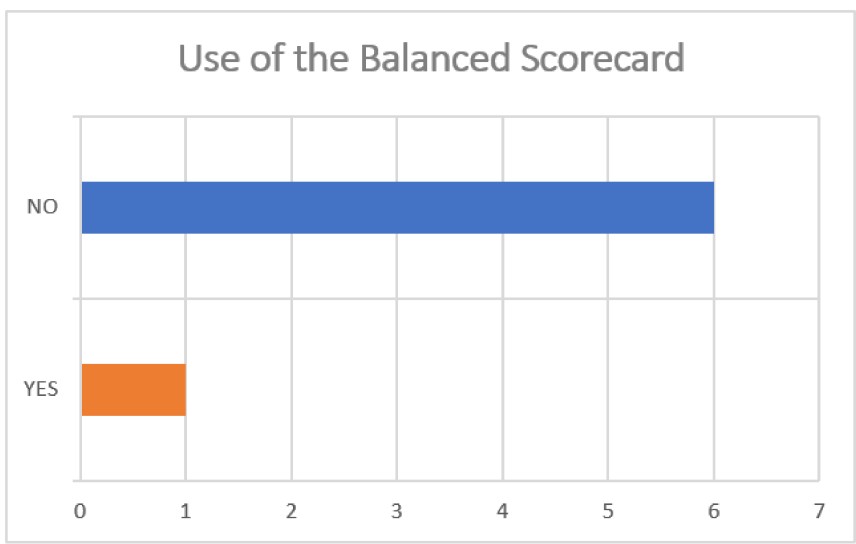

**Figure 5.** Use of the balanced scorecard.

Most of the interviewees stated that goals were established to evaluate the performance of the different areas of the company and that these were set by consensus.

Most of the interviewees stated that employees' performance is evaluated on an individual basis, although occasionally some interviewees stated that it is also done on a group and global basis. The evaluation is carried out objectively based on the results, but in some cases, in addition to the objective evaluation, it is also evaluated subjectively or qualitatively, based on the attitude, commitment, and identification of the employee with the company.

Three of the interviewees stated that they use specialized software (ERP SAP) as the information technology used to store and process data on the activities carried out in each of the company's areas and that the company's information flows through the Internet. The others stated that they were working on developing an integrated networked system. Most of the interviewees used the information from the results of their control systems for diagnostic, corrective, and proactive purposes.

Four of the interviewees stated that they did not have a system of incentives (economic and/or non-economic) to compensate for the achievement of the established goals. The rest of interviewees stated that they had a system of economic and non-economic incentives.

Most of the interviewees stated that the TBL dimensions were considered in the company's controls, although two interviewees mentioned that only the economic and social dimensions were considered. The main indicators used to measure the company's performance were focused on the economic-financial, commercial, production, and personnel control (see Table A1)

When asked which indicators they would propose to measure the social and environmental management of their company, most of the interviewees were not very explicit and limited themselves to mentioning the measurement of the work environment or the satisfaction of the personnel and the management of solid waste from the company's processes. Only the interviewee from the large company proposed, in addition, indicators to measure the company's external social management (see Table A2)

All interviewees responded that if it were demonstrated that economic results are positively correlated with social and environmental results, they would incorporate environmental and social aspects into their management control and performance measurement system. Two interviewees emphasized that it is necessary to find a way to make Peruvian industrialists aware of the need to incorporate sustainability in their management and that this would be the only way to convince the businessman that investing in social and environmental actions is self-sustainable.

## 5. Discussion and Conclusions

### 5.1. Discussion of Findings

Although most of the interviewees stated that sustainability was incorporated into their strategy, the reality is that sustainability was partially incorporated only. They prioritized the economic aspect over the social and environmental aspects to afford operating costs and generate profits. Therefore, it is necessary to demonstrate to them that investing in these last two aspects is highly correlated with the economic results, but that the returns on investment will occur in the long term and not in the short term. With respect to the social aspect, it is necessary to make them remember that, in addition to the accomplishment of the labor legislation, the enterprise must perform actions for the benefit of the community. Regarding the environmental aspect, all the interviewees placed it last in priority even though the plastics industry has become the "enemy" of the environ-mental movements. This is a profoundly serious mistake because it affects the company's image. However, it should also be noted that Peruvian market conditions often force businessmen to prioritize not the environmental aspect but rather the price, and there is also a lack of mandatory reporting of environmental actions to any regulatory body, unlike what happens in developed countries. Yet despite this, businessmen must understand that compliance with the three aspects must have equal priority because this is beneficial for society and for the conservation of the environment, upon which the survival of mankind depends.

With respect to strategic management, it is necessary for Peruvian manufacturing enterprises, especially medium-sized companies in the plastics sector, to make long-term strategic plans including sustainability and use holistic performance management tools to be competitive. Finally, it should be mentioned that an important factor to fully implement sustainability in companies is the recruitment of talented people who can carry out this transformation. It is important to remember that talent acquisition is the fourth pillar of sustainability [54].

### 5.2. Conclusions

The objective of this work was to explore the incorporation of sustainability into strategy and MCSs in Peruvian manufacturing enterprises in the plastics sector, as a case to identify and analyze the current way they incorporate and manage sustainability in selected companies to determine the shortcomings that must be corrected in the future to design an effective performance management system that incorporates sustainability to help companies achieve a competitive advantage. The main factors analyzed in this study were sustainability and the triple bottom line, the strategy, and the management control systems. The findings were obtained from four Peruvian manufacturing firms in the plastic sector through seven semi-structured in-depth interviews.

Departing from previous literature, this study adds evidence to the way that the integration of sustainability into corporate strategies has not yet been fully resolved, especially in emergent economies [5]; that relatively few methodologies and tools are applied in the manufacturing environment and that they lack a holistic approach to sustainability [35]; and that there is a considerable lack of concrete measurements, efficiency analysis, and comprehensive controlling tools that encompass economic, social, and ecological issues [36]. Besides, neither controlling practice nor sustainability are routine and commonplace but rather differ significantly at the company size [42,43], and controllers are more involved in

short-term operational tasks and functions than in long-term strategic issues [44], especially in medium-sized enterprises.

From a managerial point of view, the contribution of this work is that the empirical findings show how sustainability is really being managed in the Peruvian manufacturing companies in the plastics sector. Therefore, the results of this work will help businessmen take the respective corrective actions to fully incorporate sustainability into the strategy of the companies they manage. In addition, the proposed integrative conceptual framework that incorporates the triple bottom line dimensions into the business strategy and in the four perspectives of the balanced scorecard to build a powerful performance management system will show the Peruvian manufacturing enterprises' managers how to design broad-scope MCSs to incorporate and manage sustainability in a correct manner, to obtain competitive advantage to achieve sustainable growth.

According to the above mentioned, we conclude that this study allowed us to learn how sustainability is really managed in the Peruvian manufacturing enterprises in the plastics industry and that it is necessary for these companies not only to incorporate sustainability into their strategy but also to implement holistic broad-scope MCSs to achieve sustainable growth. Likewise, given the importance of the Peruvian manufacturing sector as a driver for growth, development, and employment, and that only 5% of Peruvian companies apply social responsibility management, we believe that the findings and proposals of this study will help Peruvian manufacturing entrepreneurs in the plastics sector to improve their knowledge and awareness about the necessity of adopting sustainable management in a correct manner to achieve the growth of their companies and the well-being of the ecosystem and society.

The limitations of the presented research study result from the small number of interviews on which the findings are based. However, it is important to highlight that this limitation is compensated by the fact that two of the interviews conducted gather the opinions of the two highest representatives of the Peruvian plastics industry guild, which strengthen and validate the results of the study. Even so, the results cannot be generalized, and further research is recommended to deepen the present findings on other types of organizations or industries, as well.

Finally, it should be mentioned that this research opens the door for future studies that may demonstrate that economic results are positively correlated with social and environmental results to convince the businessman to invest in social and environmental actions.

**Author Contributions:** Conceptualization: L.J.C.-A.; Methodology: L.J.C.-A. and J.M.R.-J.; Validation: L.J.C.-A. and J.M.R.-J.; Formal analysis: L.J.C.-A. and J.M.R.-J.; Investigation: L.J.C.-A.; Resources: L.J.C.-A.; Data curation: L.J.C.-A. and J.M.R.-J.; Writing-original draft preparation: L.J.C.-A.; Writing-review and editing: L.J.C.-A.; and J.M.R.-J.; Visualization: L.J.C.-A.; and J.M.R.-J.; Supervision: J.M.R.-J.; Project administration: L.J.C.-A.; All authors have read and agreed to the published versión of the manuscript.

**Funding:** This research received no external funding.

**Institutional Review Board Statement:** Not applicable.

**Informed Consent Statement:** Not applicable.

**Data Availability Statement:** Not applicable.

**Acknowledgments:** The authors thank the academic editor and the anonymous referees for their constructive and helpful suggestions on the early version of the paper. Likewise, we thank the Peruvian businessmen who gently let us interview them.

**Conflicts of Interest:** The authors declare no conflict of interest.

## Appendix A

Interview Questions

1. Does sustainability mean anything to you? If so, what do you understand by sustainability? Could you comment on whether sustainability has ever been considered in your company's activities and if it has not been taken into account, could you mention the reason?
2. Does TBL mean anything to you? If so, what do you understand by TBL? What actions related to social and environmental management have been developed recently in your company?
3. In the management of your company, which aspect, economic, social, or environmental, is more important to you? Why?
4. If you adopt or would adopt social or environmental actions, for what reason do you or would you do so? For ethical reasons, for economic reasons, out of obligation by any regulatory body?
5. Does the company periodically report the results of its social and environmental actions to any government regulatory body or other entity?
6. Do you have any requirements regarding social or environmental aspects from your customers and/or suppliers (domestic or foreign)?
7. Does the company have a strategic plan?
8. Do you believe that the aspect of sustainability should be taken into account in the strategy of a manufacturing company?
9. Is the sustainability aspect incorporated into your company's strategic plan (Vision, mission, objectives, values)?
10. What do you believe are the main key factors for the future success of your organization? What are the current strategic objectives of your organization?
11. Do you believe that the management of social and environmental aspects influences the economic performance of a manufacturing company? Why?
12. Do you think it is important to take into account TBL dimensions to achieve a competitive advantage? Why?
13. How is compliance with the strategic plan measured?
14. Do you know what the balanced scorecard is and do you use it?
15. If you do not use the BSC, which management control system do you use?
16. Are goals established to evaluate the performance of the different areas of the company? How are these goals set (imposition, consensus, benchmarking, etc.)?
17. How is employee performance evaluated, objectively/subjectively, individually/group/globally?
18. What type of information technology do you use to store and process the data of the activities carried out in each of the areas of the company? How does the information flow in the company (Internet/intranet, etc.)?
19. For what purposes do you use the information from the results of your control systems? Diagnostic/corrective? Interactive/proactive? Both purposes?
20. Do you have any incentive system (economic and/or non-economic) to compensate the achievement of the established goals?
21. Have the TBL dimensions been considered in the company's controls? Why?
22. What are the main indicators you use to measure the company's performance?

    – Economic—financial:
    – Customers—sales:
    – Internal processes—production/operations:
    – Personnel—IT—innovation:

23. What indicators would you propose to measure the social and environmental management of your company?

    – Social:
    – Environmental:

24. If economic performance were shown to be positively correlated with social and environmental performance, would you incorporate environmental and social aspects into your management control and performance measurement system?

## Appendix B

**Table A1.** Indicators currently used by the interviewees.

| Interview | Perspective | Indicator |
|---|---|---|
| 1 | Finance<br>Clients<br>Internal processes<br>Learning and growth | Sales volume, customer satisfaction<br>% of shrinkage, standard times |
| 2 | Finance<br>Clients<br>Internal processes<br>Learning and growth | % of shrinkage, man hours, machine hours |
| 3 | Finance<br>Clients<br>Internal processes<br>Learning and growth | Net income, acidity ratio, financial expense<br>Total sales, sales/customer, percentage change in sales by period<br>Production volume, % of shrinkage, man hours, machine hours<br>Accident rate, absenteeism rate |
| 4 | Finance<br>Clients<br>Internal processes<br>Learning and growth | ROIC, NOPAT, EBITDA<br>Customer satisfaction, customer loyalty, product quality, positioning, brand perception<br>Production volume, % of shrinkage, compliance with environmental measurements, compliance with worker health, number of hours of machine downtime, compliance with machine maintenance plan<br>Training hours, training effectiveness, organizational culture, %SAP system coverage |
| 5 | Finance<br>Clients<br>Internal processes<br>Learning and growth | Profitability by product, net income, cost of products<br>Market share, market penetration<br>Analysis of product and raw material quality variables, % of wastage, labor productivity<br>Hours of training, % of attendance, acquisition of new technology |
| 6 | Finance<br>Clients<br>Internal processes<br>Learning and growth | Return on sales, liquidity, net income<br>Sales mix, customer segmentation, customer portfolio growth, sales growth, quota compliance by salesperson (by product and by value)<br>% of defectives, % of waste, % of reprocesses, machine hours, man hours<br>attendance, overtime |
| 7 | Finance<br>Clients<br>Internal processes<br>Learning and growth | Machine hours worked/machine hours programmed, production obtained/production standard, defective units produced/units produced |

## Appendix C

**Table A2.** Social and environmental indicators proposed by interviewees.

| Interview | Perspective | Indicator |
|---|---|---|
| 1 | Social<br>Environmental | Compliance with social regulations<br>Compliance with environmental regulations |
| 2 | Social<br>Environmental | The level of well-being of personnel in the workplace<br>————- |
| 3 | Social<br>Environmental | Personnel satisfaction<br>————- |

**Table A2.** *Cont.*

| Interview | Perspective | Indicator |
|---|---|---|
| 4 | Social | Externally: Satisfaction of the communities in which work has been carried out. Links with municipalities' social assistance programs.<br>Internally: Labor climate index. |
| | Environmental | Compliance with environmental regulations. Management of material waste by the employee. |
| 5 | Social | Labor climate |
| | Environmental | ———- |
| 6 | Social | Compliance with regulations, organizational climate, reduction in overtime |
| | Environmental | Solid waste management, reduction in electricity consumption, reduction in water consumption (although water is recycled, check for water leaks) |
| 7 | Social | ———- |
| | Environmental | ———- |

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
