# Peer review of "Exploring the Inclusion of Sustainability into Strategy and Management Control Systems in Peruvian Manufacturing Enterprises"

_sustainability, doi:10.3390/su13095127_

Round 1
Reviewer 1 Report
The paper's composition is coherent; the structure is logical and meets the goal of the paper. The title "Exploring the inclusion of Sustainability into Strategy and Management Control Systems in Peruvian Manufacturing Enterprises" puts well the paper's objective; it is clear and expresses the issue being assessed very well. The results have a certain scientific value and therefore it may be interested both for journal readers as well as other professional communities. The abstract is formulated adequately along with the true picture of the paper. Conclusions are related to the results presented before reflecting the assessed issue at a professional level. I found the paper well-written and cohesive. Authors appear to be professionals, very well oriented and involved in the observed issue. The length of the paper is adequate to the significance of the topic. However major revision would be necessary to get the manuscript published in the journal. It is recommended that the authors make a relatively major revision, and the specific amendments to the text are as follows:
- In Introduction part I recommend mentioning the way how the research results could be implemented in the practice bringing up any benefits and added value by expressing the research novelty.
- The Conclusion section is recommended to be set aside from the Discussion section. Some kind of polemic discourse comparing the research outcomes with the literature overview part would be beneficial to be involved in Discussion.
- In Conclusions section it is recommend mentioning the way how the research results could be implemented in the practice bringing up any benefits and added value by expressing the research novelty.
- As the ranking of Sustainability journal is pretty high, more sophisticated statistical methods and assessments (e.g. correlation analysis) would be necessary to be implemented in the paper including hypothesis estimation. Some charts reflecting the questionnaire outputs would be appreciated as well.
Author Response
Dear Reviewer,
Thank you very much for your comments. They have contributed to improve and complete our paper.
Following your recommendations:
- In the introduction section, paragraph 11, we have included the work’s value.
- Section 5: Discussion and conclusions; has been divided into two sections: 5.1. Discussion of findings, and 5.2: Conclusions.
- In the conclusions section (Section 5.2, paragraph 3), it has been specified that the proposed conceptual framework (See Fig 1) has been designed to address the shortcomings found in the research and its implementation will help Peruvian entrepreneurs to achieve sustainable growth in their companies. In addition, in section 5.1 (last paragraph), it is emphasized that to fully implement sustainability in companies, it is necessary for these companies to acquire talented people.
- Regarding the last observation, we should mention that we have carried out a qualitative study, so we have not used sophisticated statistical tools. But following your valuable recommendation, we have included some graphs (See Figs., 2,3,4 and 5) that reflect the results of the questionnaire of the interviews carried out.
Sincerely yours,
The authors.
Reviewer 2 Report
The following sentences are very long, obscure and need to be rephrased for better clarity.
- >>"In addition, it should be mentioned that it is crucial for organizations the incorporation of sustainability in its strategy not only
for not destabilizing the natural systems we rely on for survival but
also for generating competitive advantage." - "For all it was previous mentioned, a thorough understanding
of the current sustainability adoption and management in Peruvian
manufacturing enterprises will serve to propose the implementation of methodologies and tools for moving from today’s most economical focus towards all three dimensions of sustainability" - "Challenges for contemporary controlling practice are to link
the larger variety of sustainability-oriented strategic goals to operations, to measure sustainability and to evaluate the achievement
of corporate goals at a high level of uncertainty but with a limited
scope of action"
Another few very complex statement not making clear sense, and MUST be reworded as they are critical/crucial points in the paper that is not making much sense:
- "This is the gap that the present study pretends to close to contribute to foster the sustainability to be adopted for the Peruvians entrepreneurs quickly and in a widespread manner because it is of vital necessity for the businesses, the planet, and the society."
- "The proposed
system embeds the three aspects of the TBL in the Vision, Mission,
Values and Strategic Objectives of the company, and leverages the
potential of the BSC to manage and control the accomplishment of
the strategic objectives showing in a clear way, through the Strategic Map, how the strategic objectives are interconnected to each
other through cause-effect relationships to achieve economic benefits in a sustainable manner."
Many more such language related grammar, punctuations and sentence structure must be improved.
Ref#26 appears in quotes, but the page number is missing. Is this expected/accepted format for the referencing style?
The idea of reviewing the balance score card is a good approach as it is considered seminal, however, it would be better to also include some more recent models related to sustainability to increase the currency of the research work along with Otley's from 1999. Eg: Try to include a review of the model proposed by Rozario et all from 2020, related to the organisation sustainability - Rozario, Sophia Diana, et al. "Enabling Corporate Sustainability from a Talent Acquisition Perspective." Journal of Sustainability Research 2.2 (2020). Some more of such recent research and models must be included in the literature review.
>> "On the other hand, performance assessment is one of the essential parts of sustainable management, because sustainability assessment is a crucial means to promote sustainable development." This transition is very abrupt and does not appear seemlessly flowing in a coherent manner from the previous para.
The first illustrations needs labelling and better explanation for clarity.
4.1. Sustainability and Triple Bottom Line - This section and many parts of the paper is very heavily skewed towards a review of 'plastic' and its part in the sustainability instead of Sustainability into Strategy and Management Control Systems as the title states.
More graphical illustration to understand the responses from the interviewees and the impact on that response can be considered.
Author Response
Dear Reviewer,
Thank you very much for your comments. They have contributed to improve and complete our paper.
Following your recommendations, we have rewrite or removed some sentences:
- Now: >>"In addition, it should be mentioned that it is crucial for organizations the incorporation of sustainability in its strategy for generating competitive advantage." (Section 1, paragraph 3)
- It was removed.
- Now: >>“Challenges for contemporary controlling practice are to link the sustainable strategic goals to operations, to measure sustainability and to evaluate the achievement of corporate goals” (Section 2.2, paragraph 6)
- It was removed.
- Now: >>"The proposed system embeds the three aspects of the TBL in the Vision, Mission, Values and Strategic Objectives of the company, and leverages the
potential of the BSC to manage and control the accomplishment of
the strategic objectives to achieve economic benefits in a sustainable manner." (Section 2.3, paragraph 3)
- We have redone the list of references considering the order of appearance in the text (previously it was in alphabetical order), reference 26 is now reference 1, and the page number has been included. (Section 1, paragraph 1)
- We have considered Rozario's recommendations (Section 5.1., last paragraph). Thank you for sharing this reference.
- Now: >> “However, even though sustainability assessment is a crucial means to promote sustainable development, relatively few methodologies and tools are applied in the manufacturing environment and they lack a holistic approach to sustainability”. (Section 2.2, paragraph 5)
- Illustration 1 was labeled and its contents were explained in greater detail for clarity. (Section 2.3., Figure 1)
- Following your valuable recommendation, we have included some graphs (See Figs., 2,3,4 and 5) that reflect the results of the questionnaire of the interviews carried out.
Sincerely yours,
The authors.
Reviewer 3 Report
Dear Authors,
Thank you for the interesting paper although I found some problems in the following parts of your paper:
First more related to the theory:
in your paper there is nothing mentioned about Sustainable Development Goals (SDGs) - does it mean that companies do not care about them?
You wrote: "Findings suggest that all the interviewees are familiar with the term sustainability and understand its importance, but they do not know how to implement it accurately" - does it mean they do not have a specified strategy? compare with: https://www.researchgate.net/publication/332072298_The_analysis_of_strategy_types_of_the_renewable_energy_sector
Your paper is actually a multiple case study analysis - please develop the description of the plastic sector and companies which are subject of your study
Other problems with your paper
Introduction:
The first sentence is missing a source indication.
There is a sentence “we need a just transformation in the way we live, work, and cooperate”. This sentence is a quotation but you have not provided any source.
dots "." should appear after source brackets not before them => [numbers].
The tables and figures are not in the format proposed for the Sustainability Journal.
Figure 1 and Table 2 have not indicated sources. Figure 1 is not titled nor elaborately described - what are the implications coming from this figure?
Also, references are not provided by the citation manager (Mendeley) what makes their format weird. In the text, it is not acceptable to put so huge collections of references [multiple numbers indicating sources].
References section:
In my opinion, the reference section is missing the recent article and positions from the Sustainability journal:
https://www.mdpi.com/2071-1050/12/21/8832
https://www.mdpi.com/2071-1050/13/4/1604
References format is not the same as established in the template
Author Response
Dear Reviewer,
Thank you very much for your comments. They have contributed to improve and complete our paper.
Following your recommendations:
- The strategic objectives set out in the proposed PMS are fully aligned with Goal 9 (Industry, innovation and infrastructure) of the SDGs. Besides, we are sure that most of the Peruvian businessmen are not aware of the SDGs, considering that only 5% of Peruvian companies apply social responsibility management. We are confident that the content of this article will contribute to the fulfillment of the SDGs, which are at country level. Implicitly, the Human Development Report, which we had mentioned, aims to achieve the SDGs.
- Thank you for sharing the reference. Respect to the stages of company greening, it could be said that the medium enterprises are in the stage of technology improvement and the large enterprise is just starting to embrace the green strategy. It was included in Section 4.2., last paragraph.
- Indeed, our paper is a multiple case study analysis. This has been corrected. (See Abstract and Section 3, first paragraph)
- It was corrected. See the first sentence of the Introduction.
- It was corrected. See Introduction, first paragraph.
- It was corrected, dots appear after source brackets. See all the document.
- It was corrected. The tables and figures are in the format proposed for the Sustainability Journal.
- Sources of Figure 1 and Table 2 are now indicated.Figure 1 was labeled, and its contents were explained in greater detail for clarity. (Section 2.3., Figure 1)
- We have redone the list of references considering the order of appearance in the text (previously it was in alphabetical order), as established in the template.
Sincerely yours,
The authors.
Round 2
Reviewer 1 Report
All the necessary observations and comments have been incorporated into the revised manuscript in a proper way giving the paper higher added value and professional features;
Reviewer 2 Report
This version is better than the previous version, there is improvement in the paper. It is worth giving another read of the paper to look for opportunities to simplify any complex and long sentences.
Reviewer 3 Report
Please check each section carefully.
Check your figures and tables layout to confirm them right.